# *Astragalus polysaccharides* (PG2) Enhances the M1 Polarization of Macrophages, Functional Maturation of Dendritic Cells, and T Cell-Mediated Anticancer Immune Responses in Patients with Lung Cancer

**DOI:** 10.3390/nu11102264

**Published:** 2019-09-20

**Authors:** Oluwaseun Adebayo Bamodu, Kuang-Tai Kuo, Chun-Hua Wang, Wen-Chien Huang, Alexander T.H. Wu, Jo-Ting Tsai, Kang-Yun Lee, Chi-Tai Yeh, Liang-Shun Wang

**Affiliations:** 1Division of Hematology & Oncology, Department of Medicine, Shuang Ho Hospital, Taipei Medical University, New Taipei City 235, Taiwan; 16625@s.tmu.edu.tw; 2Department of Medical Research and Education, Shuang Ho Hospital, Taipei Medical University, New Taipei City 235, Taiwan; 3Division of Thoracic Surgery, Department of Surgery, Shuang Ho Hospital, Taipei Medical University, New Taipei City 235, Taiwan; doc2738h@gmail.com; 4Division of Thoracic Surgery, Department of Surgery, School of Medicine, College of Medicine, Taipei Medical University, Taipei City 110, Taiwan; 5Department of Dermatology, Taipei Tzu Chi Hospital, Buddhist Tzu Chi Medical Foundation, New Taipei City 231, Taiwan; 10205@s.tmu.edu.tw; 6School of Medicine, Buddhist Tzu Chi University, Hualien City 970, Taiwan; 7Department of Medicine, MacKay Medical College, New Taipei City 252, Taiwan; wjhuang0@yahoo.com.tw; 8Division of Thoracic Surgery, Department of Surgery, MacKay Memorial Hospital, Taipei City 104, Taiwan; 9The Ph.D. Program for Translational Medicine, College of Medical Science and Technology, Taipei Medical University, Taipei City 110, Taiwan; chaw1211@tmu.edu.tw; 10Department of Radiation Oncology, Shuang Ho Hospital, Taipei Medical University, New Taipei City 235, Taiwan; kitty4024@gmail.com; 11Department of Radiology, School of Medicine, College of Medicine, Taipei Medical University, Taipei City 110, Taiwan; 12Division of Pulmonary Medicine, Department of Internal Medicine, Shuang Ho Hospital, Taipei Medical University, New Taipei City 235, Taiwan; leekangyun@tmu.edu.tw; 13Department of Biotechnology and Pharmaceutical Technology, Yuanpei University of Medical Technology, Hsinchu City 30015, Taiwan

**Keywords:** *Astragalus polysaccharide*, PG2, cisplatin, macrophages, monocytes, M1/M2 polarization, immunotherapy, maintenance therapy, lung cancer, NSCLC

## Abstract

Background: Recently, we demonstrated that *Astragalus polysaccharide* (PG2), the active ingredient in dried roots of *astragalus membranaceus,* ameliorates cancer symptom clusters and improves quality of life (QoL) in patients with metastatic disease by modulating inflammatory cascade against the background roles of inflammatory cells, including macrophages, dendritic cells (DCs), and cytotoxic T lymphocytes (CTLs) in tumor initiation, metastasis, and progression. Nevertheless, the role of PG2 in the modulation of anticancer immunogenicity and therapeutic response remains relatively underexplored and unclear. Purpose: The present study investigates how and to what extent PG2 modulates cellular and biochemical components of the inflammatory cascade and enhances anticancer immunity, as well as the therapeutic implication of these bio-events in patients with lung cancer. Methods and Results: Herein, we demonstrated that PG2 significantly increased the M1/M2 macrophage polarization ratio in non-small cell carcinoma (NSCLC) H441 and H1299 cells. This PG2-induced preferential pharmacologic up-regulation of tumoral M1 population in vitro positively correlated with the downregulation of tumor-promoting IL-6 and IL-10 expression in NSCLC cell-conditioned medium, with concomitant marked inhibition of cell proliferation, clonogenicity, and tumorsphere formation. Our ex vivo results, using clinical sample from our NSCLC cohort, demonstrated that PG2 also promoted the functional maturation of DCs with consequent enhancement of T cell-mediated anticancer immune responses. Consistent with the in vitro and ex vivo results, our in vivo studies showed that treatment with PG2 elicited significant time-dependent depletion of the tumor-associated M2 population, synergistically enhanced the anti-M2-based anticancer effect of cisplatin, and inhibited xenograft tumor growth in the NSCLC mice models. Moreover, in the presence of PG2, cisplatin-associated dyscrasia and weight-loss was markedly suppressed. Conclusion: These results do indicate a therapeutically-relevant role for PG2 in modulating the M1/M2 macrophage pool, facilitating DC maturation and synergistically enhancing the anticancer effect of conventional chemotherapeutic agent, cisplatin, thus laying the foundation for further exploration of the curative relevance of PG2 as surrogate immunotherapy and/or clinical feasibility of its use for maintenance therapy in patients with lung cancer.

## 1. Introduction

Lung cancer is one of the most frequently diagnosed malignancies and the leading cause of cancer death worldwide, with an estimated 234,030 new cases diagnosed and 154,050 deaths in the United States alone in 2018 [1]. Non-small cell lung cancer (NSCLC) accounts for about 85% of all lung cancers, and its current five-year survival remains below 20% [1]. The touted therapeutic success of targeted therapies, such as documented improvement in tumor response rates and longer progression-free survival [2] in the last decade, has been limited by very narrow therapeutic selectivity and large non-eligible patient pools, thus necessitating the discovery of a novel therapeutic agent with broad spectrum anticancer effects and efficacy. Recently, compared with conventional chemotherapy, immune checkpoint inhibitors, especially those targeting programmed cell death protein 1 (PD-1), have been shown to effectively prolong the overall survival (OS) of patients with recurrent and advanced NSCLC [3,4,5]. However, despite this reported success of the immune checkpoint inhibitors in various cancers, the response rate is very modest, with less than 20% of patients responding to them [6]. This dismal response rate may be connected with the complex tumor microenvironment (TME), and the contextual deleterious influence of immune suppressor cells in the non-responders. 

In the context of oncogenic cues from a dysregulated TME, evidenced abounds that dendritic cells (DCs), in their capacity as nature’s adjuvant, and professional antigen-presenting cells (APCs) play a central role in triggering anticancer immunity against cancerous cells; thus, the rationale for DC therapy and resultant induction of T cell-based anticancer immunogenic response [7]. Therefore, any therapy that modulates the pro-inflammatory/anti-inflammatory cell balance manipulates these cells towards enhanced tumoricidal immunogenicity without fanning the flames of oncogenicity, and abrogate acquired immune privilege would constitute a novel therapeutic agent with high efficacy in patients with cancer [8,9,10,11].

Accumulated evidence indicates that inflammatory cells significantly influence the TME either by their direct cell-to-cell effects, or indirectly through the production of cytokines and chemokines, and these secreted mediators of inflammation play specific roles in regulating tumor cell proliferation, angiogenesis, and metastasis [12,13,14]. While the immunogenic role of tumor-associated macrophages (TAMs) is well documented, an increased TAM population has also been associated with poor prognosis for patients with cancer. This may be linked to temporospatial variation in the phenotypical status of tumor-resident or associated macrophages. Phenotypically, macrophages are broadly divided into two biological classes, namely classical M1 and alternative M2 macrophages [15,16]. The M1 macrophages driven by the Th1 cytokine interferon-γ, bacterial moieties, and Toll-like receptor (TLR) agonists, produce pro-inflammatory cytokines, such as Interleukin (IL)-6, IL-12, IL-23, and tumor necrosis factor-α (TNF-α), and are involved in inflammatory responses and anticancer immunity [17]. Conversely, the M2 macrophages are associated with Th2 cytokines, such as IL-4, IL-10, IL-13, and are implicated in anti-inflammatory responses and tumor-promoting signals [18,19,20]. Importantly, besides being a useful predictor of clinical outcomes, the reprogramming of the TAMs from M2 to M1 phenotype is of therapeutic relevance, especially in terms of anticancer immunotherapy. 

*Astragalus polysaccharide* (PG2), the active ingredient from dried roots of *astragalus membranaceus* (Chinese: *Huang-Chi*), exhibits a broad spectrum of documented therapeutic effects, including immune modulation [21,22]. However, the immune-dependent anticancer activity of PG2 in lung cancer remains largely unexplored and unclear. Thus, following our recent works demonstrating that PG2 ameliorates cancer-related fatigue and other cancer symptom clusters, as well as improving the quality of life (QoL) of patients with metastatic disease through the modulation of patients’ inflammatory cascade [23,24] against the background that macrophages and other immune cells; playing critical roles in tumor initiation, dissemination, and progression; as well as suggestions that PG2 may play a role in the differentiated status of splenic DCs; increased CD11c^high^CD45RB^low^ DC population; and enhanced immune function of T lymphocyte via up-regulation of the Th1/Th2 ratio while in vitro [25], the present study hypothesized and demonstrated that PG2, by preferentially up-regulating the M1/M2 macrophage ratio, skewing the pro-inflammatory/anti-inflammatory signaling balance towards an anticancer phenotype, and repressing erstwhile-acquired immune privilege by cancerous cells in patients with NSCLC.

## 2. Materials and Methods

### 2.1. Chemicals and Reagents

*Astragalus polysaccharide* (PG2) lyophilized powder obtained from PhytoHealth (PhytoHealth Corporation, Taipei, Taiwan), and cisplatin purchased from Sigma-Aldrich (St Louis, MO, USA), were dissolved in dimethyl sulfoxide (DMSO) to generate 10mM stock solutions. Verapamil, Hoechst 33342, and methylthiazolyldiphenyl-tetrazolium bromide (MTT) dye were also purchased from Sigma-Aldrich (St. Louis, MO, USA). Primary antibodies against CD80, CD206, NF-κB p65, CD11b, CD31, IL-4, IL-6, IL-10, IL-13, Interferon gamma (IFN-γ), and β-actin were purchased from Cell Signaling Technology (Boston, MA, USA), while human recombinant IL-4, IL-13, IFN-γ, granulocyte-macrophage colony stimulating factor (GM-CSF), and lipopolysaccharide (LPS) were obtained from R&D Systems, Inc. (Minneapolis, MN, USA).

### 2.2. Peripheral Blood Mononuclear Cells (PBMCs) Culture and Isolation of Dendritic Cells

The present study was approved by the research ethics and procedures institutional review board of the Taipei Medical University (Approval no.: 2018-IRB-0027). After obtaining informed consent, peripheral blood samples were drawn from patients with lung cancer (*n* = 17). After PBMCs isolation, 1 × 10^6^ PBMCs were seeded per mL of complete cell growth medium supplemented with 10% fetal calf serum (FCS) per well of 96-well deep round plates in 5% humidified CO_2_ atmosphere, at 37 °C, overnight. Thereafter, the PBMCs were transferred into 10 mL tissue culture dishes at a final volume of 7 mL and incubated in the presence or absence of 25 ng/ml phorbol 12-myristate 13-acetate (PMA) for 24 h, 20 ng/ml IL-4 and IL-13 for 24 h, 20 ng/ml LPS and INF-γ for 24 h, or 16 mg/mL PG2 for 48 h, in 5% CO_2_ humidified atmosphere at 37 °C. Percentage of CD80+ or CD206+ macrophages was then determined by flow cytometry. The study started in November 2012 and was completed in June 2017 (clinical trial information: IRB No.: 201205017 and NCT01720550).

### 2.3. Cell Lines and Culture

The human lung cancer H441 (ATCC HTB-174), H1299 (ATCC CRL-5803), H1437 (ATCC CRL-5872), and murine Lewis Lung cancer (LLC1, ATCC CRL-1642) cell lines used in this study were obtained from the ATCC (American Type Culture Collection, Manassas, VA, USA) and cultured in Roswell Park Memorial Institute (RPMI)-1640 (Invitrogen, San Diego, CA, USA) or Dulbecco’s modified Eagle’s medium (DMEM, Gibco-Life Technologies Inc., Gaithersburg, MD, USA) supplemented with 10% heat-inactivated fetal bovine serum (FBS, Invitrogen), 100 UI/mL penicillin, and 100 μg/mL streptomycin at 37 °C in humidified 5% CO_2_ atmosphere. Cells were sub-cultured or used when they attained ≥80% confluence. The human monocytic THP-1 cells (ATCC TIB-202; American Type Culture Collection, Manassas, VA, USA) were cultured in RPMI-1640 (Invitrogen) supplemented with 10% heat-inactivated FBS (Invitrogen), 10 mM HEPES (Gibco-Life Technologies Inc., Gaithersburg, MD, USA), 2.5 g/L D-glucose (Merck Millipore, Jaffrey, NH, USA), 1 mM pyruvate (Gibco-Life Technologies Inc., Gaithersburg, MD, USA), and 50 pM β-mercaptoethanol (Gibco-Life Technologies Inc., Gaithersburg, MD, USA). 

THP-1 monocytes were differentiated into macrophages by incubating the THP-1 cells in 25 ng/mL PMA (Sigma-Aldrich Corporation, St. Louis, MO, USA) for 24 h, followed by incubation in RPMI-1640 for another 24 h. The monocyte-derived macrophages (MDMs) were polarized into M1 macrophages by incubation with 20 ng/ml IFN-γ (R&D system, Minneapolis, MN, USA) and 10 pg/ml LPS (Sigma-Aldrich Corporation, St. Louis, MO, USA), while M2 macrophages were generated by incubating the MDMs in 20 ng/ml IL-4 and IL-13 (R&D Systems, Minneapolis, MN, USA). 

For co-culture assays, MDMs seeded onto six 0.4 μm-pored Transwell membrane inserts (Corning Inc., Corning, NY, USA). The MDMs and lung cancer cell lines were co-cultured in RPMI-1640 medium supplemented with 0.5 mM L-glutamine (Sigma-Aldrich Corporation, St. Louis, MO, USA) and 3.75 g/L D-glucose (Sigma) for 24 h before incubation, with or without PG2 and/or cisplatin. The retrieved supernatant from culturing the THP-1 cells in PMA (Sigma) was used as monocyte-conditioned medium (MCM).

### 2.4. Western Blot

After lysing the cells in Radio-Immunoprecipitation Assay (RIPA) lysis buffer (Pierce, Rockford, IL, USA) supplemented with protease inhibitor cocktail and phenylmethylsulfonyl fluoride (PMSF; Sigma-Aldrich Corporation, St. Louis, MO, USA), 20μg protein lysate samples were loaded per lane into 10% sodium dodecyl sulfate polyacrylamide gel electrophoresis (SDS-PAGE) gels (Bio-Rad); after electrophoresis, the separated protein blots were transferred onto nitrocellulose membranes (Millipore Corporation, Bedford, MA, USA), membranes were incubated with 5% skimmed-milk in Tris-buffered saline with Tween 20 (TBST) for 1 h to block non-specific binding, and then incubated overnight at 4 °C with primary antibody against CD80 (1:1000, #15416, Cell Signaling Technology, Inc., Danvers, MA, USA), CD206/MRC1 (1:1000, #91992, Cell Signaling Technology, Inc., Danvers, MA, USA), or β-actin (1:1000, #3700, Sigma-Aldrich Corporation, St. Louis, MO, USA), washed in TBST three times and then incubated in horseradish peroxidase (HRP)-conjugated goat anti-rabbit (1:5000) or anti-mouse (1:3000) secondary antibody (Millipore Corporation, Bedford, MA, USA). Thereafter, Pierce enhanced chemiluminescence (ECL) western blotting substrate (Pierce) was used for band detection and imaging done with the BioSpectrum Imaging System (UVP, Upland, CA, USA).

### 2.5. Flow Cytometry Analyses

After staining the THP-1 cells in an Invitrogen LIVE/DEAD Fixable Red Dead Cell Stain kit (#L23102, Thermo Fisher Scientific Inc., Waltham, MA, USA), viable cells were blocked using 10μg/mL human immunoglobulin G (IgG) diluted in 0.5% bovine serum albumin (BSA)/1mM, ethylenediaminetetraacetic acid (EDTA)/0.01%, sodium azide staining buffer on ice for 10 min, and then incubated with anti-human CD206 (MMR)-PE antibody (#12-2069-41, 1:50, eBioscience, Thermo Fisher Scientific Inc., Waltham, MA, USA) on ice for 1 h, before cell fixation in 2% paraformaldehyde (PFA) on ice for 15 min. Next, the cells were incubated with anti-human CD80 (B7-1) Fluorescein isothiocyanate (FITC) antibodies (#11-0809-41, 1:50, eBioscience, Thermo Fisher Scientific Inc., Waltham, MA, USA) in 0.5% saponin on ice for 45 min, washed with staining buffer, and then flow-cytometry data acquisition was performed using the BD FACSCanto II system (BD Biosciences, Thermo Fisher Scientific Inc. Waltham, MA, USA). Data obtained were analyzed using Flowjo software v. 9.6.2 (FlowJo, LLC, Tree Star, Inc., Ashland, OR, USA). 

### 2.6. Methylthiazolyldiphenyl-Tetrazolium Bromide (MTT) Viability Assay

After the differentiation of THP-1 cells seeded at 1.8 × 10^5^ cells/well in 24-well plates into MDMs, as already described above, the MDMs were incubated in IFN-γ and LPS, then in 500 μl MTT reagent for 2 h at 37 °C in a humidified 5% CO_2_ atmosphere incubator. The MCM was carefully removed, 1mL lysis buffer was added to the MDM cells in each well, cells were gently agitated at 70 rpm for 1 h at 37 °C, and then cell-viability-related absorbance was read at a wavelength of 450 nm using a SpectraMax i3x microplate reader (Molecular devices, Kim Forest Enterprises Co., Ltd., New Taipei City, Taiwan).

### 2.7. Cell Migration and Invasion Assays

To assess migration potential, NSCLC cells were cultured to ≥90% confluence in 6-well plates, and then a scratch was made along the median axis of the monolayer confluent cells using sterile yellow pipette tips. Phosphate-buffered saline (PBS) was used to carefully wash off detached cells and cellular debris, and the scratch-wound closure monitored over indicated time. The mean number of cells that migrated into the denuded area from five randomly selected visual fields was determined in each dish for comparison of wound closure at indicated time-points. 

For invasion assay, matrigel-coated transwell inserts with 8 m-pored membranes (BD Biosciences, San Jose, CA, USA) were placed in 24-well plates. Then, 3 × 104 cells were seeded into 100 μl FBS-free media in the upper chambers and allowed to invade through the membrane into the lower chamber containing 600 μl complete growth medium, with 10% FBS serving as chemoattractant. Cells were incubated in 5% CO_2_ humidified atmosphere at 37 °C for 48 h. The non-invaded cells were carefully scraped from the upper surface of each insert with sterile cotton buds, while the invaded cells on the lower surface of the insert membrane were incubated in 0.1% crystal violet at 37 °C for 30 minutes, washed twice with 1X PBS, and viewed under a microscope.

### 2.8. Colony Formation Assay

H441 or H1299 cells were seeded at a density of 1.5 × 10^4^/well in 6-well plates (Corning Inc., Corning, NY, USA) containing in complete growth medium, and incubated for 12 days in humidified 5% CO_2_ atmosphere at 37 °C. Formed colonies with a diameter ≥100 μm were photographed and counted from five randomly selected non-overlapping visual fields under microscope after staining with 0.01% (w/v) crystal violet for 45 min.

### 2.9. Enzyme-Linked Immunosorbent Assay (ELISA)

Cell culture media after cell stimulation and/or treatment were assayed for secreted IL-6 (ab178013) and IL-10 (ab185986) cytokine using ELISA kits (Abcam plc., Cambridge, MA, USA) following the manufacturer’s instructions. Absorbance was measured at 450nm using an iMark microplate absorbance reader (Bio-Rad Laboratories, Inc., Hercules, CA, USA), and the concentration of the secreted cytokines was calculated using standard curves.

### 2.10. Tumor Implantation and Growth in Syngeneic Mice Models

Six to eight week-old female C57BL/6 mice (*n* = 15, median weight: 24.5 ± 1.8 g) were purchased from the National Laboratory Animal Center (NARlabs, Taipei City, Taiwan) and housed in the MacKay Memorial Hospital animal research facility under specific pathogen-free (SPF) conditions. All animal procedures and experimental methods were compliant with the National Institutes of Health guidelines for the care and use of laboratory animals [26] and approved by the Animal Care and User Committee at MacKay Memorial Hospital (Affidavit of Approval of Animal Use Protocol # MMH-LAC-107-001). One group of C57BL/6 mice (*n* = 10) were inoculated with 1.5 × 10^6^ LLC1 cells, while another set (*n* = 5) were inoculated with 1 × 10^6^ H1437 cells and 5 × 10^5^ THP-1 cells in 20 μl PBS intraperitoneally. The size of ensuing tumors was measured weekly for 17 weeks using calipers, and tumor volume (*v*) was calculated using the formula *v* (mm^3^) = [*l* × *w*^2^]/2, where *l* is the length of tumor longest diameter and *w* is the length of shortest diameter. Once tumor volume of 200 mm^3^ was attained, the mice were divided into treatment and control groups randomly. PG2 (3 mg/kg/day) or vehicle (0.1% DMSO) was injected biweekly for 16 weeks, intraperitoneally. The mice were then humanely sacrificed, and lung tumor tissues were excised and analyzed.

### 2.11. Ethics Approval and Consent to Participate 

The study started in November 2012 and was completed in June 2017 (Clinical trial information: IRB No.: 201205017 and NCT01720550). All animal procedures and experimental methods were compliant with the National Institutes of Health guidelines for the care and use of laboratory animals (National Academies Press (U.S.), 2011) and approved by the Animal Care and User Committee at MacKay Memorial Hospital (Affidavit of Approval of Animal Use Protocol # MMH-LAC-107-001).

### 2.12. Statistical Analysis

All data are representative of the mean ± standard error of the mean (SEM) of assays performed at least three times in triplicates. Statistical analysis was performed using the Statistical Package of Social Sciences software (IBM Corp. Released 2016. IBM SPSS Statistics for Windows, Version 24.0. Armonk, NY: IBM Corp.). Comparison between the 2 groups was done using a two-sided Student’s *t*-test, while for comparison between ≥3 groups, the one-way analysis of variance (ANOVA) was used. A *p*-value < 0.05 was considered significant statistically.

## 3. Results

### 3.1. Macrophages Respond Differentially to Different Inflammatory Cytokine Stimuli

After differentiation of the PMA-exposed human THP-1 monocytes into macrophages (MDMs), and stimulation with IFN-γ/LPS or IL-4/IL-13 for macrophage polarization (Figure 1A), immunofluorescence staining and flow cytometry was used to confirm the functional phenotype of the resultant cells based on expression of established macrophage markers, CD80 and CD206 for M1- and M2- polarized macrophages, respectively. A significant increase from 0.813% to 62.3% in the CD80+ M1 population was noted after IFN-γ/LPS activation (Figure 1B, upper panel), while the IL-4/IL-13 stimulation induced an increase from 0.817% to 52.8% in the CD206+ M2 macrophages (Figure 1B, lower panel). Because of previous studies implicating TAMs in the acquisition and maintenance of cancer stem cell (CSCs)-like phenotypes [17,27], we examined the probable effects of M1- or M2-conditioned media on H1299 side population and CSCs-like phenotypes. We observed that the M2-conditioned medium derived from the supernatant from the IL-4/IL-13 exposed culture significantly increased the side population after 48 h of incubation, compared to the M1 conditioned medium group (M2 vs. M1: 2.97-fold increase, *p* < 0.05) (Figure 1C). Moreover, compared with the H1299 cells treated with M1-conditioned medium, exposure to M2-conditioned medium elicited a significantly greater number of invaded cells by the 18 h time-point (Figure 1D).

### 3.2. PG2 Enhances M1 Polarization While Down-Regulating IL-4/IL-13-Induced M2 Polarization Dose-Dependently

To gain some mechanistic insight into the therapeutic activities of PG2 observed in our recent works [23,24] against the background of its documented immunomodulatory activities [21,22,26], we further investigated the effect of PG2 on the M1/M2 macrophage skewness in NSCLC cell line H441 co-cultured with MDMs. As demonstrated in our results, compared to the PMA-treated control group, the population of CD206+ M2-polarized MDMs increased from 0.044% to 66.2% after exposure to IL-4/IL-13, while the CD80+ M1 macrophages increased from 0.589% to 1.59% (Figure 2A,B). In parallel assays, 48 h exposure to 16 mg/mL PG2 significantly expanded the CD80+ M1-polarized MDM pool from 1.59% pre-treatment level to 78.1%. Conversely, 16 mg/mL PG2 markedly reduced the CD206+ M2 cells (51.32-fold, *p* < 0.01) (Figure 2C). Halving the dose of PG2 (8 mg/mL) elicited similar skewness of the M1/M2 population howbeit to a lesser degree (Figure 2D). 

### 3.3. The Enhancement of M1 Macrophage Polarization by PG2 Is Akin to the Effect of LPS/IFN-γ Stimulation of MDMs

For translational relevance, we further compared the M1-promoting activity of PG2 with that of the classical LPS/IFN-γ-stimulated M1 polarization. Firstly, PMA-differentiated M1- and M2-polarized MDMs (Figure 3A) were treated with LPS/IFN-γ or grown in cancer cell culture medium (CCCM) obtained from prior culturing of H1299 or H441 cells. As expected, exposure to 40 ng/mL LPS and 200 ng/mL IFN−γ enhanced the population of CD80+ M1 MDMs by 62.6 % (*p* < 0.05) with corresponding 0.860 % (*p* < 0.05) increase in CD206+ M2 MDMs (Figure 3B). Comparatively, exposure of the polarized MDM population to CCCM elicited 17.1% (*p* < 0.05) and 61.9% (*p* < 0.01) increase in M1 and M2 macrophages, respectively (Figure 3C). Interestingly, subsequent treatment of the CCCM-treated MDMs with 16 mg/mL PG2 for 48 h elicited a 67.9% increase in CD80+ M1 and ~58% reduction in CD206+ M2 cells (Figure 3D,E). These results indicate that the induction of M1 polarization by PG2 treatment is akin to that elicited by LPS/IFN-γ stimulation, and that PG2 treatment effectively reverses the CCCM-induced M2-skewed macrophage pool with preferential enhancement of the M1/M2 ratio. 

### 3.4. PG2 Represses the Tumor-Promoting Effects of Anti-Inflammatory Cytokines and Inhibits the NSCLC Stem Cell-Like Phenotypes Induced by M2-Conditioned Medium

In separate experiments, we assessed if, how, and to what extent PG2-modulated CD80+M1/CD206+M2 ratio affects the CSCs-like phenotypes of NSCLC. Consistent with earlier results, western blot analyses showed that while PG2 up-regulated the expression level of CD80 protein, it down-regulated CD206 protein expression (Figure 4A). Furthermore, we demonstrated that PG2 significantly repressed anti-inflammatory IL-10-induced proliferation of tumor-promoting CD206+ cells, compared with the control group (~2.58-fold, *p* < 0.01), and this was akin to the PG2-induced 2.93-fold (*p* < 0.01) decrease in CD206 expression by M2 macrophages cultured in CCCM (Figure 4B). In validation assays using ELISA, we demonstrated that the significantly enhanced production of IL-10 (2.21-fold, *p* < 0.001) and IL-6 (3.33-fold, *p* < 0.001) by M2 macrophages was efficiently repressed on exposure to 16mg/mL PG, as reflected by observed 2.05-fold (*p* < 0.01) and 3.16-fold (*p* < 0.01) decrease in IL-10 and IL-6, respectively (Figure 4C,D). In the light of these findings, we next examined the effect of PG2 on the CSCs-like phenotype, using in vitro NSCLC stem cell models. Interestingly, we observed that in contrast to the cultured M1-MCMs which remained largely adherent, cultured M2-MCMs, to a large extent, acquired an anchorage-independent phenotype and increased ability to form tumorspheres, which are in vitro CSCs models (Figure 4E, upper panel). This increased tumorsphere formation ability was more apparent in the H1299 cells. In addition, we observed that co-culture with M1-MCMs induced a moderate decrease in the viability and/or proliferation of the H1299 (I.27-fold, *p* < 0.01) and H441 (I.56-fold, *p* < 0.01) cells, compared to the M2-co-cultured groups (Figure 4E, lower panel). Of therapeutic relevance, treatment with 0 – 16 mg/mL PG2 significantly suppressed the viability and disrupted the morphological consistency of formed tumorspheres from the M2-co-cultured H1299 cells in a dose-dependent manner (Figure 4F). These data suggest a putative role for PG2 as an effective inhibitor of M2 macrophage polarization and CSCs-phenotype acquisition in vitro.

### 3.5. PG2 Suppresses Tumorigenicity and Metastasis in Syngeneic C57BL/6 Mice, and Potentiates Anticancer Effect of Cisplatin In Vivo by Modulating Inflammation-Associated Macrophage Activity and Angiogenesis

In addition, we evaluated the combinatorial potential of PG2 with current chemotherapy by comparatively analyzing the pharmacological effect of PG2 and/or cisplatin, a first line chemotherapeutic agent for NSCLC [28,29] in syngeneic LLC1 tumor-bearing C57BL/6 mice. We demonstrated that while cisplatin or PG2 monotherapy reduced the size and weight of tumors formed in the syngeneic mice over 17 weeks, with a 48.5% (*p* < 0.001) or 53.4% (*p* < 0.001) decrease in tumor size at week 17, respectively, cisplatin/PG2 combination caused a 74.5% (*p* < 0.001) reduction in tumor size (*p* < 0.001) (Figure 5A,B). Of clinical relevance, we observed that while cisplatin caused significant loss of mice body weight (*p* < 0.001), no apparent weight loss was seen in mice treated with PG2 alone or in combination with cisplatin (Figure 5B). More interestingly, we observed that compared to the vehicle (DMSO)-treated mice with 71% appreciable metastasis nodules, treatment with cisplatin reduced the number of metastasis nodules by 39.3% (*p* < 0.05), while the PG2-treatment and PG2/cisplatin combination elicited 75.4% (*p* < 0.01) and 91.8% (*p* < 0/01) reduction in the number of metastasis nodules counted (Figure 5C). These data indicate that PG2 synergistically enhanced the anticancer effect of cisplatin, and that PG2 alone or in combination effectively suppressed tumor growth and spontaneous lung metastasis in NSCLC mice models in vivo.

To gain some mechanistic insight into the observed PG2 pharmacological activities, we examined its effect on the cancer-complicit component of the inflammatory cascade, the nuclear factor kappa protein, beta subunit (NF-κB) [30], pro-angiogenic marker CD31, and pro-inflammatory macrophage marker CD11b [31]. Using tumor samples obtained from the mice, we demonstrated that treatment with 0.5 mg/kg/day cisplatin alone, 3 mg/kg/day PG2 alone, or PG2/cisplatin combination down-regulated the expression of NF-κB and CD31 proteins mildly, moderately, and strongly, respectively, while up-regulating CD11b protein expression level in increasing order of magnitude (Figure 5D). These results further support the tumor-suppressing effect of PG2 and its ability to enhance the anticancer effect of cisplatin.

### 3.6. PG2 Up-Modulates the CD80+ M1/CD206+ M2 Macrophage Ratio and Increases the Population of CD80+, CD103+, and CD86+ Functionally Matured Dendritic Cells Ex Vivo

To further characterize the immunogenic potential of PG2, using ELISA and flow cytometry, we comparatively analyzed the ability of PMA, LPS+INF-γ, or PG2 to modulate the differentiation of the M1/M2 macrophage pool into dendritic cells (DCs) and the functional maturation of immature DCs (iDCs) to mature DCs (mDCs) in cancer patients. We observed that compared to PMA, LPS+INF-γ or PG2 induced a 2.4-fold (*p* < 0.01) and 3.7-fold (*p* < 0.01) increase in the number of CD80+ M1 macrophages, respectively, with PG2 inducing 8.5% more CD80+ M1 than LPS+INF−γ (Figure 6A). Conversely, PG2 reduced the number of CD206+ M2 macrophage by 12.1% (*p* < 0.05) and 6.0% (*p* = not significant) compared to PMA and IL-4/IL-13, respectively (Figure 6B). Since granulocyte-macrophage colony-stimulating factor (GM-CSF) and interleukin-4 (IL-4) trigger monocyte differentiation into DCs and prevent differentiation to osteoclasts (OCs), as well as induce DC differentiation in the presence of macrophage colony-stimulating factor (M-CSF) and receptor activator of nuclear factor kappa-B (RANK) ligand [32], we compared the effect of GM-CSF+IL4 and PG2 on monocyte differentiation to DCs and DC maturation. We demonstrated that compared to treatment with GM-CSF+IL4 alone, exposure to 16 mg/mL PG2 alone significantly increased the population of CD80+, CD103+, and CD86+ cells by 7% (*p* < 0.05), 8% (*p* < 0.05), and 1% (*p* = not significant), respectively (Figure 6C,D,E), while compared to GM-CSF+IL4 alone, GM-CSF+IL4+PG2 induced more CD80+ (16%, *p* < 0.01), CD103+ (15%, *p* < 0.01), and CD86+ (13%, *p* < 0.05) cells (Figure 6C,D,E). These data indicate, at least in part, that PG2 facilitates T cell activation, M1 differentiation to DCs, and maturation of DCs to mDCs in patients with NSCLC. 

To determine the ‘reproducibility’ and ‘generalizability’ of these data, we probed for similar trends in patients with breast (*n* = 4), colon (*n* = 2), ovarian (*n* = 2), liver (*n* = 3), gastric (*n* = 2) and brain (GBM, *n* = 2) cancers. Our results showed that PG2 alone or in combination with GM-CSF+IL4 also enhanced the production of functional mDCs in the breast, colon, ovarian, liver, gastric, and brain cancers (Figure 6F and Appendix A). Moreover, in light of PG2 immunogenicity in most malignancies, and the observed ability of PG2 to modulate the DCs and macrophage-associated immune landscape, we further investigated if and to what extent treatment with PG2 would affect the activation status and type of T cells in tumor niche. Results of our flow cytometry analysis of dissociated cell suspensions from patient-derived tumor samples indicated that 16 mg/kg PG2 significantly increased the population of CD45+ leucocytes/white blood cells (WBCs) (3.02-fold, *p* < 0.001), which coincidentally seems to be due to a significant 9.4% expansion of the CD45+CD8+ T cells (*p* < 0.01), with no apparently significant variation in the number of CD4+ T cells in PG2-treated tumor samples compared to their untreated counterparts (Appendix A). 

## 4. Discussion

Corroboratory to documentation of PG2 potently modulating immunity in in vivo inflammation [33] and diabetes [34] models, our recent works demonstrated that PG2 ameliorates cancer-related fatigue and other cancer symptom clusters, as well as improves the quality of life (QoL) of patients with metastatic disease through the modulation of patients’ inflammatory cascade [23,24]; however, the underlying molecular or cellular mechanisms remain relatively unclear. While PG2 is known to exhibit anti-inflammatory potentials, as demonstrated by its ability to enhance the expression of anti-inflammatory cytokines, such as interleukin (IL)-10, transforming growth factor (TGF)-β, and 5′ adenosine monophosphate- activated protein kinase (AMPK), as well as inhibit expression of pro-inflammatory cytokines, including IL-1β, inducible nitric oxide synthase (iNOS), monocyte chemoattractant protein (MCP)-1, IL-6, and CD11c in an AMPK-dependent manner [35], it has also been suggested that PG2 can promote the inflammatory process by increasing the level of pro-inflammatory cytokines, such as tumor necrosis factor (TNF)-α, granulocyte-macrophage colony-stimulating factor (GM-CSF), NF-κB, and the production of nitric oxide (NO) [36]. Such contradictory findings suggest the diversity of PG2 functions, and our current work was aimed to elucidate the role of PG2 in macrophage-modulatory activity more clearly. In this study, using the MDM experimental model, we induced M1- and M2-polarized macrophages by exposure to type I cytokines IFN-γ and LPS, or type II cytokines IL-4 and IL-13, respectively (Figure 1). This differential response of macrophages to different inflammatory cues is consistent with contemporary knowledge on molecular determinants of macrophage polarization [18] and makes room for data replicability.

Pursuant to accruing indications of the probable therapeutic benefit of manipulating the immune-modulating potential for immune-based anticancer therapy, we demonstrated, for the first time and to the best of our knowledge, that akin to LPS/IFN-γ stimulation, PG2 dose-dependently enhances M1 polarization while down-regulating IL-4/IL-13-induced M2 polarization (Figure 2 and Figure 3). This ability of PG2 to modulate tumoral M1/M2 polarization with preferential enhancement of M1 polarization is of clinical relevance, considering current experimental and clinical evidence indicating that M1 macrophages suppress tumorigenicity, decrease the viability and proliferation of cancer cells, and enhance the sensitivity of cancer cells to chemotherapy [37,38]. Additionally, two other studies, regardless of using host-produced histidine-rich glycoprotein (HRG) or phospholipase D4 (PLD4), have also implied the potential utility of M1 polarization as an effective anticancer therapeutic tool [39,40]. Thus, this PG2-induced depletion of M2 MDMs constitutes a potential new modality in immune-based anticancer therapy for patients with NSCLC.

Furthermore, consistent with the broadly known well-knit interaction between the cellular and biochemical components of the inflammatory cascade [17,27,41], we also demonstrated that concomitant with documented enhancement of M1 polarization at the expense of a depleted M2 pool, PG2 markedly represses the tumor-promoting effects of anti-inflammatory cytokines and inhibits the NSCLC stem cell-like phenotypes induced by M2-conditioned medium (Figure 3 and Figure 4). The pro-inflammatory cytokine IL-6 is of relevance in lung cancer biology, pathology, and therapy, especially as it interacts with and activates the signal transducer and activator of transcription (STAT)-3, which is constitutively activated in various types of malignancies [41]. In fact, IL-6/STAT3 signaling has been shown to suppress lung cancer initiation while promoting cancerous cell proliferation, migration, and disease progression by positively modulating the induction of cyclin D1 [41]. We thus posit that the PG2-induced downregulation of IL-6 and IL-10 suggests that, dependent on the TME and cytokine landscape, PG2 exhibits not only a tumoricidal but also tumoristatic effects on NSCLC cells.

Moreover, our data showing that PG2 elicits inhibition of NSCLC stem cell-like phenotypes induced by M2-conditioned medium (Figure 4 and Appendix A) is therapeutically relevant, considering CSCs exhibit unlimited self-renewal potential, enhanced ability to drive tumor initiation and disease progression, as well as well-documented complicity in therapy failure and disease recurrence [16,27,36,42]. Beyond being passive residents of the tumor niche, TAMs/MDMs induce autocrine and paracrine signals that enhance CSC survival, maintenance, self-renewal, motility, and tumorigenicity [27]. In addition, TAM-mediated induction of epithelial-to-mesenchymal transition (EMT), shown to be functionally relevant to the acquisition and maintenance of the CSC-like phenotype, does suggest the critical role of TAMs in the acquisition of a metastatic phenotype by the CSCs [42]. This highlights the therapeutic relevance of PG2 as a potential CSC-targeting pharmacological agent and putative component of an effective anticancer therapeutic strategy for patients with NSCLC.

Furthermore, we demonstrated that PG2 suppresses tumorigenicity and metastasis in syngeneic C57BL/6 mice, and potentiates cisplatin anticancer effect in vivo, by modulating inflammation-associated macrophage activity and angiogenesis (Figure 5). Despite being touted as a potent anticancer agent across a broad spectrum of malignancies, the therapeutic benefit of cisplatin is significantly limited by the cancerous cells’ innate or acquired resistance upon exposure to cisplatin. Our results are consistent with reports associating reduced cisplatin efficacy or ‘cisplatin resistance’ with NF-κB activation and/or aberrant expression [43]. Kuang et al. [43] demonstrated that the activation of NF-κB is essential for cisplatin-resistance, as reflected by up-regulated expression of anti-apoptosis proteins and attenuated cell death in cisplatin-resistant nasopharyngeal carcinoma cells compared to the cisplatin-sensitive cells, thus suggesting that combining cisplatin with a NF-κB inhibitor, such as PG2, may help enhance the therapeutic efficacy of cisplatin; thus, the significance of our data shows that PG2 suppresses the nuclear accumulation of NF-κB and synergistically enhances the anticancer effect of cisplatin (Figure 5).

Similarly, CD31/platelet endothelial cell adhesion molecule (PECAM)-1, has been implicated in the pathogenesis of several malignancies. Corroborating our findings showing that PG2 suppresses NF-κB and CD31/PECAM-1 (Figure 5), there is evidence that CD31 enhances transendothelial migration of leukocytes and up-regulates NF-κB expression and/or activity, while CD31low/null endothelial cells exhibit defective induction of Phosphatidylinositol 3-kinase (PI3K)/Akt, mitogen-activated protein kinase (MAPK)/extracellular signal-regulated kinase (ERK), and Src signaling, as well as loss of NF-κB-induced gene transcription [44,45]. Thus, the putative clinical implication of PG2-induced CD31 inhibition in reduced tumor neovascularization, altered vascular permeability, enhanced susceptibility of endothelial cells to apoptotic stress, and attenuated intratumor leukocyte infiltration. In fact, this is supported by our demonstrated association of PG2 treatment with enhanced CD11b expression (Figure 5), especially as CD11b+ TAMs/MDMs display of an anti-cancer phenotype and association with reduced angiogenesis, reversed immune suppression, acquisition of a benign phenotype, repressed invasiveness, and invariably disease remission [30,46,47]. 

Finally, in ex vivo studies, consistent with the previously demonstrated role of PG2 in the modulation of the functional status and polarization of T cells in mice with polymicrobial sepsis [33], we demonstrated, for the first time to the best of our knowledge, that PG2 modulates the CD80+ M1/CD206+ M2 macrophage pool and increases the population of functional CD80+CD103+CD86+ mDCs derived from PBMCs of patients with different cancer types (Figure 6). This of particular clinical relevance because as natural adjuvants and professional antigen-presenting cells (APCs), DCs play a vital immune-related anticancer role, as demonstrated by the use of DC therapy to stimulate T cell-based anticancer response [7], especially with its ability to cross-present antigens with immune-stimulating potential, enhance the MHC class I-mediated MHC class II presentation of exogenous antigens, and enable direct stimulation of CD8+ T cells for anticancer immune responses. Meanwhile, considering that not all mDCs are immunogenic with anticancer potential, our data showing that PG2 increases the systemic pool of functional CD80+CD103+CD86+, mDCs also bear therapeutic relevance, as CD80, CD83, CD86, and CD103 have been reported as biomarkers of functionally matured DCs with enhanced anticancer potential [48,49]. 

## 5. Conclusions

Taken together, these results demonstrate, as shown in graphical abstract Figure 7, for the first time the novel role of PG2 as an effective modulator of tumoral M1/M2 macrophage polarization and a potent activator of DC maturation. These results provide some mechanistic insight into observed macrophage-mediated, PG2-induced, immune-based anticancer and chemotherapy-potentiating activity. This study thus lays the foundation for further exploration of the role of PG2 as alternative cancer immunotherapy or maintenance therapy for patients with NSCLC.

## Figures and Tables

**Figure 1 nutrients-11-02264-f001:**
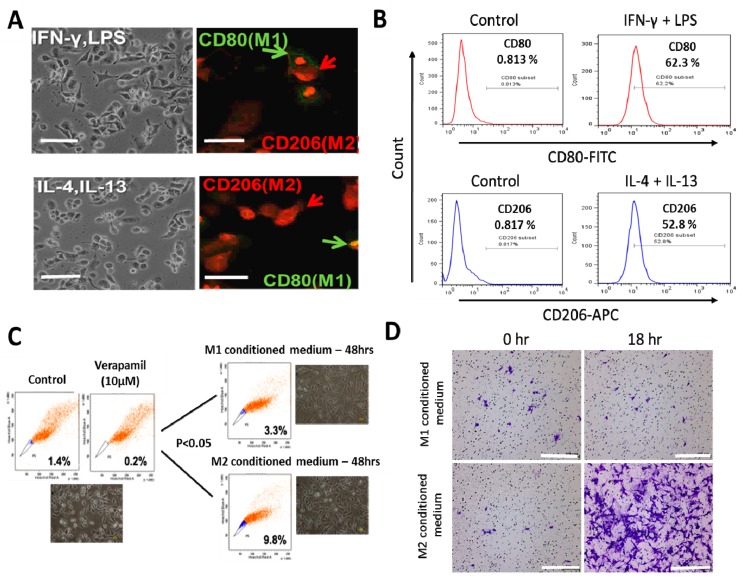
Differential macrophage response to different inflammatory cytokine stimuli. (**A**) Morphological and immunocytochemical images showing monocyte-derived macrophage (MDM) polarization to M1 or M2 functional phenotype using type 1 cytokine IFN−γ and lipopolysaccharide (LPS) or type 2 cytokine IL-4 and IL-13, respectively. (**B**) Increases in CD80+ M1 and CD206+ M2 MDMs were seen after treatment with IFN−γ /LPS or IL-4/IL-13, respectively, using flow-cytometry analysis. Green arrows point to the CD80+ M1 macrophage, while red arrows indicate CD206+ M2 macrophages. (**C**) Flow cytometry and morphology imaging of M1 and M2 cell sorting and isolation using the fluorescence-activated cell sorting (FACS) assay. (**D**) Compared with the H1299 cells treated with M1 conditioned medium, the H1299 cells cultured with M2 conditioned medium exhibited higher invasion ability at 18 h in matrigel study. APC: antigen-presenting cell.

**Figure 2 nutrients-11-02264-f002:**
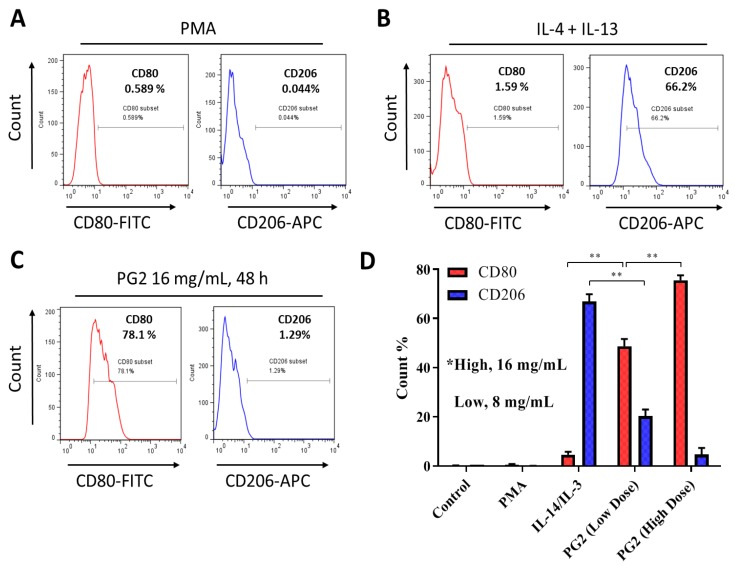
*Astragalus polysaccharide* (PG2) enhances M1 polarization and down-regulates IL-4/IL-13-induced M2 polarization. Images from flow cytometric analyses showing (**A**) the differentiation of THP-1 monocyte into macrophages after 24 h incubation in phorbol 12-myristate 13-acetate (PMA), (**B**) incubation of MDM in IL-4 and IL-13 induced a CD206^high^CD80^low^ M2 phenotype, while (**C**) incubation of MDM in PG2 16 mg/mL for 48 h induced a CD80^high^CD206^low^ M1 phenotype. (**D**) A graphical representation of the differential effect of IL-4/IL-13 and low dose (8 mg/mL) or high dose (16 mg/mL) PG2 treatment on M1–M2 polarization. ***p* < 0.01

**Figure 3 nutrients-11-02264-f003:**
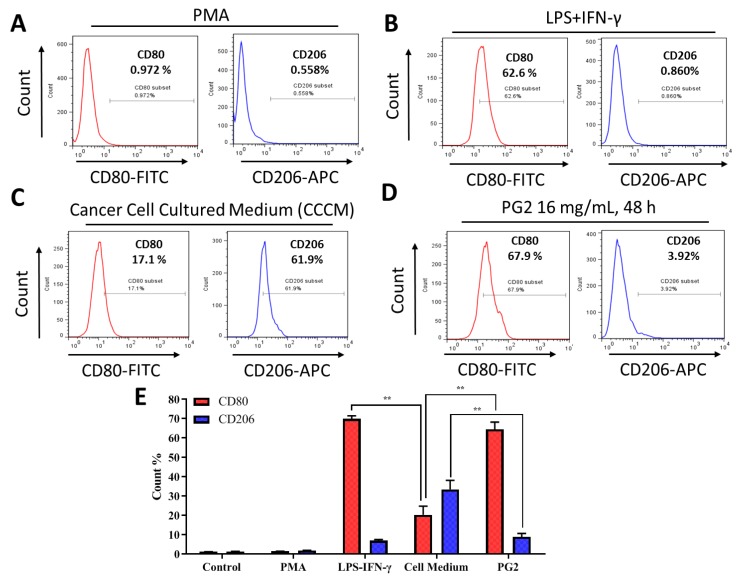
The enhancement of M1 macrophage polarization by PG2 is akin to the effect of LPS/IFN-γ stimulation of MDMs. Images from flow cytometric analyses showing (**A**) the differentiation of THP-1 monocyte into macrophages after 24 h incubation in PMA; (**B**) MDMs after exposure to IFN-γ and LPS induced a CD80^high^CD206^low^ M1 phenotype; (**C**) cancer cell culture medium (CCCM) induced 17.1% CD80+ and 61.9% CD206+ MDMs; (**D**) PG2-treatment of MDMs pre-incubated in CCCM induced a CD80^high^CD206^low^ M1 phenotype, similar to IFN-γ/LPS exposure; (**E**) a graphical representation of the differential effect of IFN-γ/LPS, CCCM, and 16 mg/mL PG2 treatment on M1–M2 polarization. PG2 enhanced the M1 phenotype akin to IFN-γ/LPS exposure effect. ***p* < 0.01

**Figure 4 nutrients-11-02264-f004:**
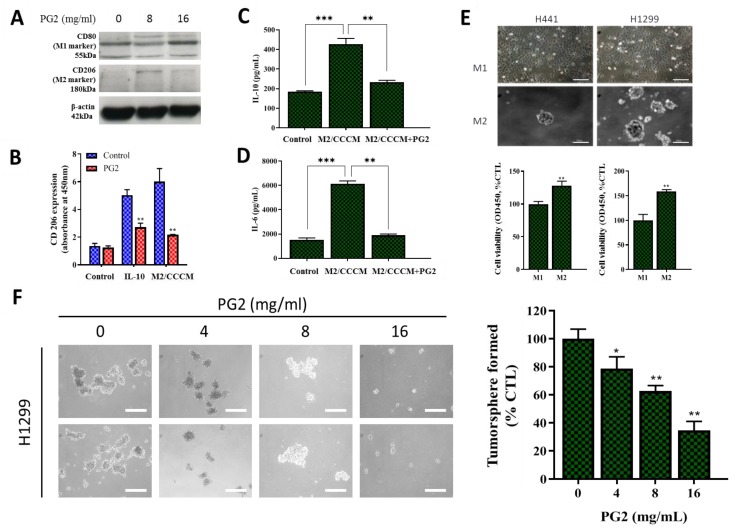
PG2 negatively modulates the secretion of tumor-promoting anti-inflammatory cytokines and inhibits the M2-MCM-induced cancer stem cell-like phenotype of NSCLC cells. (**A**) Western blot showing a dose-related up-regulation of CD80 and down-regulation of CD206 protein expressions in MDMs after PG2 treatment. (**B**) Representative histogram of ELISA result showing PG2 significantly inhibited IL-10-enhanced M2 macrophage proliferation and CCCM-cultivated M2 population, compared to the untreated control group. ELISA histogram showing that 16 mg/mL PG2 exposure significantly reversed the CCCM-enhanced (**C**) IL-10 and (**D**) IL-6 secretion by M2 cells. (**E**) Photo (*upper panel*) and graphical (*lower panel*) images showing that H441 and H1299 cells co-cultured with M1-MCM remained adherent, while M2-MCM co-cultured cells acquired an adhesion-independent cancer stem cell (CSC) phenotype. (**F**) PG2 inhibited the viability of tumorspheres grown from M2/H1299 co-culture in a dose-dependent manner. **p* < 0.05, ***p* < 0.01, ****p* < 0.001; β-actin served as loading control.

**Figure 5 nutrients-11-02264-f005:**
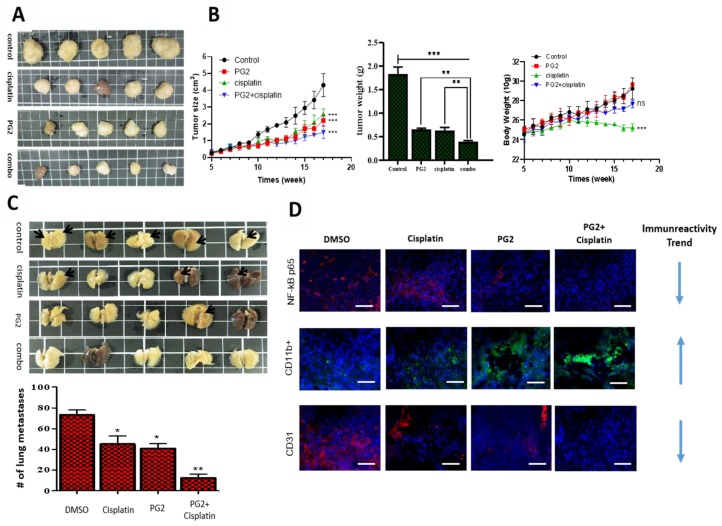
PG2 suppresses tumorigenicity and metastasis in syngeneic C57BL/6 mice and potentiates cisplatin anticancer effect in vivo by modulating inflammation-associated macrophage activity and angiogenesis. (**A**) Photo images show the anticancer effect of cisplatin and/or PG2 in syngeneic C57BL/6 mice inoculated with 1.5 × 10^3^ LLC1 cells. (**B**) Graphical representation of the effect of cisplatin and/or PG2 on the tumor size, tumor weight, and body weight in syngeneic C57BL/6 mice inoculated with LLC1 cells. (**C**) Photo images show the effect of cisplatin and/or PG2 on metastasis in syngeneic C57BL/6 mice inoculated with LLC1 cells. (**D**) Immunofluorescent staining showed that PG2 or cisplatin alone mildly reduced the expression of beta subunit (NF-κB), CD11b, and CD31, while combining cisplatin with PG2 significantly inhibited their expression in the tissue specimen. Red arrow, liver metastasis; ns, not significant; **p* < 0.05, ***p* < 0.01, ****p* < 0.001; DMSO, dimethyl sulfoxide

**Figure 6 nutrients-11-02264-f006:**
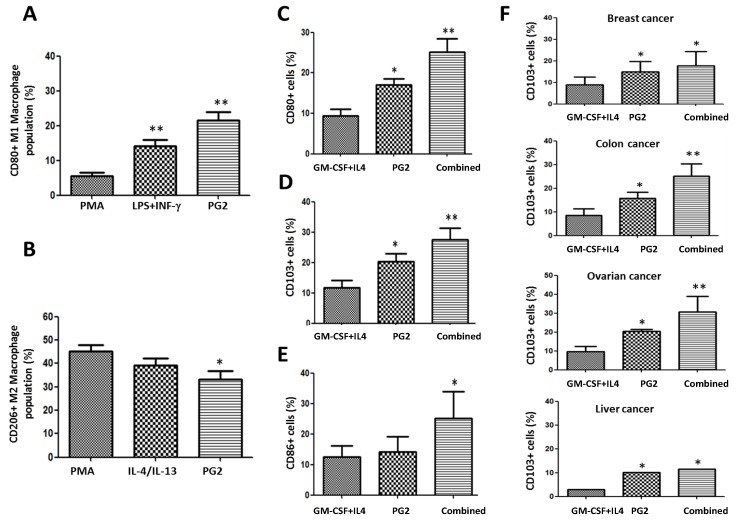
PG2 modulated the CD80+ M1/CD206+ M2 macrophage population and increased the population of CD80+, CD103+, and CD86+ dendritic cells derived from peripheral blood mononuclear cells (PBMCs) of cancer patient’s ex vivo. The effect of PMA, LPS+INF-γ, or PG2 on the proportion of (**A**) CD80+ and (**B**) CD206+ cells, as shown by flow cytometry. Graphical representation of the effect of PG2 on the population of (**C**) CD80+, (**D**) CD103+, and (**E**) CD86+ dendritic cells derived from GM-CSF+IL-4-treated PBMCs of NSCLC patients. (**F**) Graphical representation of the effect of PG2 on the number of functional CD103+ dendritic cells derived from GM-CSF+IL-4-treated PBMCs of breast, colon, ovarian, or liver patients. 1: GM-CSF+IL4; 2: GM-CSF+IL4, followed by the treatment of PG2 (16 mg/mL); 3: GM-CSF+IL4, washed out, and followed by the treatment of PG2 (16 mg/mL); **p* < 0.05, ***p* < 0.01.

**Figure 7 nutrients-11-02264-f007:**
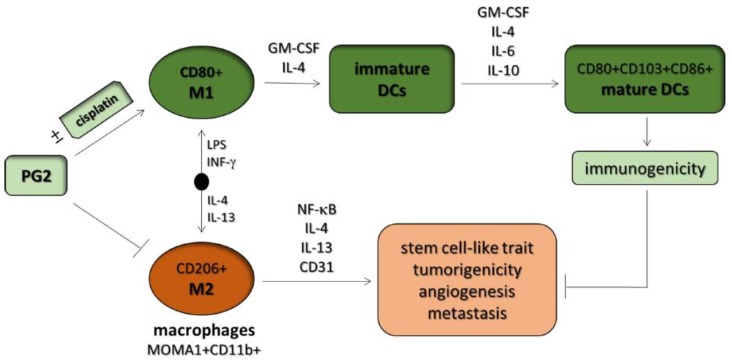
Graphical abstract. *Astragalus* polysaccharide (PG2) enhances the phenotypic polarization of macrophages, functional maturation of dendritic cells, and T cell-mediated immune responses for anticancer therapy. DCs: dendritic cells, GM-CSF: granulocyte-macrophage colony stimulating factor.

## Data Availability

The datasets used and analyzed in the current study are available from the corresponding authors in response to reasonable requests.

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
