# Peer review of "Astragalus polysaccharides* (PG2) Enhances the M1 Polarization of Macrophages, Functional Maturation of Dendritic Cells, and T Cell-Mediated Anticancer Immune Responses in Patients with Lung Cancer"

_nutrients, 2019, doi:10.3390/nu11102264_

Round 1
Reviewer 1 Report
The present study reveals novel immunomodulatory roles attributed to Astragalus polysaccharide (PG2) in vitro. Overall, the study shows that PG2 reverses the population of macrophages from M2 to M1, reduces the cytokine production and stemness of lung cancer cells induced by M2 macrophages, decreases tumor growth and lung metastases in vivo in combination with cisplatin, and promotes the maturation of dendritic cells from PBMCs from patients with different types of cancer. Given the data provided herein, PG2 could represent an attractive approach to enhance or maintain the tumor response to immunotherapy that would deserve future clinical trials.
However, there are several aspects that authors must improve/modify in the manuscript:
- The authors state that PG2 enhances T cell-mediated anticancer immune response (title) and facilitates T cell activation (page 12, line 430) based on the effects of PG2 on the maturation of dendritic cells. However, there is not any experiment that supports that affirmation. The authors should either show changes (enhancement) in T cell populations, or activation, by flow cytometry or modify the title and delete that affirmation.
- The authors also state in the title and different parts of the entire manuscript that the study was done in patients with lung cancer. This assertion is not real as the experiments were done in commercially available monocytes, cell lines, mice and cancer-patient-derived PBMCs. This terminology is confusing because it means that any experimentation was made in human patients. It should be modified in the entire manuscript.
- In Materials and Methods, the number of patients with other types of cancer (breast, colon, ovarian liver, gastric and brain cancers) must be included.
- Results on cell migration assays are not in the manuscript. Please, include them.
-Figure 1C and D about the impact on side population and invasion should be in a separate figure or in figure 4.
- Figure 1D: the picture of M1 and M2 conditioned media at 0h are the same. Please, provide the correct images.
- Figures 2 and 3. Blue and red colors were used to refer to CD206 and CD80, respectively, in the flow cytometry histograms. However, in the graphs, colors are used in the opposite direction. Please, unify criteria between histograms and bar graphs to avoid confusion.
- Figure 4F should be done in the H441 cell line. Besides, the authors must indicate whether that experiment was made in co-culture with M2 macrophages, conditioned media, or in raw parental cells. If not done, this experiment should be done in M2-macrophages conditioned media.
- Figure 5C. Show representative immunohistochemistry images of lung metastases in each treatment group.
Author Response
Answers to the comments:
Point-by-point responses to reviewer’s comments:
We would like to thank the reviewer for the thorough reading of our manuscript as well as their valuable comments. We have followed their comments closely and feel that they have further improved the readability and appeal of our work, as well as strengthened the manuscript. Below are our point-by-point responses.
Q1: Reviewer #1: The present study reveals novel immunomodulatory roles attributed to Astragalus polysaccharide (PG2) in vitro. Overall, the study shows that PG2 reverses the population of macrophages from M2 to M1, reduces the cytokine production and stemness of lung cancer cells induced by M2 macrophages, decreases tumor growth and lung metastases in vivo in combination with cisplatin, and promotes the maturation of dendritic cells from PBMCs from patients with different types of cancer. Given the data provided herein, PG2 could represent an attractive approach to enhance or maintain the tumor response to immunotherapy that would deserve future clinical trials. However, there are several aspects that authors must improve/modify in the manuscript.
A1: We sincerely thank the reviewer for the time taken to review our work, the words of encouragement, and for the important suggestion made to improve the quality of our work.
Q2: The authors state that PG2 enhances T cell-mediated anticancer immune response (title) and facilitates T cell activation (page 12, line 430) based on the effects of PG2 on the maturation of dendritic cells. However, there is not any experiment that supports that affirmation. The authors should either show changes (enhancement) in T cell populations, or activation, by flow cytometry or modify the title and delete that affirmation.
A2: A2: We appreciate the reviewer’s insightful comment. As requested by the reviewer, we have now included data indicating that PG2 does enhance T cell-mediated anticancer immune response and facilitates T cell activation. Please kindly see our revised Results section, Lines 412-446.
3.6. PG2 up-modulates the CD80+ M1/ CD206+ M2 macrophage ratio and increases the population of CD80+, CD103+, and CD86+ functionally matured dendritic cells, ex vivo
To further characterize the immunogenic potential of PG2, using ELISA and flow cytometry, we comparatively analyzed the ability of PMA, LPS+INF-g or PG2 to modulate the differentiation of the M1/M2 macrophage pool into dendritic cells (DCs) and the functional maturation of immature DCs (iDCs) to mature DCs (mDCs) in cancer patients. We observed that compared to PMA, LPS+INF-g or PG2 induced a 2.4-fold (p<0.01) and 3.7-fold (p<0.01) increase in the number of CD80+ M1 macrophages, respectively, with PG2 inducing 8.5% more CD80+ M1 than LPS+INF-g (Fig. 6A). Conversely, PG2 reduced the number of CD206+ M2 macrophage by 12.1% (p<0.05) and 6.0% (p not significant) compared to PMA and IL-4/IL-13, respectively (Fig. 6B). Since granulocyte-macrophage colony-stimulating factor (GM-CSF) and interleukin-4 (IL-4) trigger monocyte differentiation into DCs, and prevent differentiation to osteoclasts (OCs), as well as induce DC differentiation in the presence of M-CSF and RANK ligand (Hiasa et al., 2009), we compared the effect of GM-CSF+IL4 and PG2 on monocyte differentiation to DCs and DC maturation. We demonstrated that compared to treatment with GM-CSF+IL4 alone, exposure to 16 mg/mL PG2 alone significantly increased the population of CD80+, CD103+ and CD86+ cells by 7% (p<0.05), 8% (p<0.05), and 1% (p = not significant), respectively (Figs. 6C, Figs. D and Figs. 6E), while compared to GM-CSF+IL4 alone, GM-CSF+IL4+PG2 induced more CD80+ (16%, p<0.01), CD103+ (15%, p<0.01) and CD86+ (13%, p<0.05) cells (Figs. 6C, Figs. D and Figs. 6E). These data indicate, at least in part, that PG2 facilitates T-cell activation, M1 differentiation to DCs, and maturation of DCs to mDCs in patients with NSCLC.
To determine the ‘reproducibility’ and ‘generalizability’ of these data, we probed for similar trends in patients with breast (n = 4), colon (n = 2), ovarian (n = 2), liver (n = 3), gastric (n = 2) and brain (GBM, n = 2) cancers. Our results showed that PG2 alone or in combination with GM-CSF+IL4 also enhances the production of functional mDCs, in the breast, colon, ovarian liver, gastric and brain cancers (Fig. 6F and Supplementary Figure S1). Moreover, in light of PG2 immunogenicity in most malignancies, and the observed ability of PG2 to modulate the DCs and macrophage-associated immune landscape, we further investigated if and to what extent treatment with PG2 would affect the activation status and type of T cells in tumor niche. Results of our flow cytometry analysis of dissociated cell suspensions from patient-derived tumor samples indicated that 16 mg/kg PG2 significantly increased the population of CD45+ leucocytes/white blood cells (WBCs) (3.02-fold, p<0.001), which coincidentally seem due to a significant 9.4% expansion of the CD45+CD8+ T cells (p<0.01), with no apparently significant variation in the number of CD4+ T cells in PG2-treated tumor samples compared to their untreated counterparts (Supplementary Figure S2).
Also kindly see our Updated Supplementary Figure S2 and its legend, please see line 720-724.
Supplementary Figure S2. PG2 treatment elicits significant increase in tumour-infiltrating CD8+ T cells population. Representative flow cytometry data (left panel) and histograms (right panel) showing the effect of PG2 on the mean (A) CD45+ leukocyte, and (B) CD8+ or CD4+ T cell population in disassociated patient-derived cell suspensions compared to untreated control tumour samples. *p<0.05, **p<0.01, ***p<0.001, ns, not significant.
Q3: Reviewer #1: The authors also state in the title and different parts of the entire manuscript that the study was done in patients with lung cancer. This assertion is not real as the experiments were done in commercially available monocytes, cell lines, mice and cancer-patient-derived PBMCs. This terminology is confusing because it means that any experimentation was made in human patients. It should be modified in the entire manuscript.
A3: We sincerely thank the reviewer for this important comment. We agree with the reviewer that our “experiments were done in commercially available monocytes, cell lines, mice and cancer-patient-derived PBMCs”; where the cancer-patient-derived PBMCs were derived from our cohort of patients with lung cancer. Thus, the assertion made is real, as we always strive towards good science, which includes data integrity and ‘reproducibility’. Please kindly see Abstract section, Lines 35-61.
Abstract:
Background: Recently we demonstrated that Astragalus polysaccharide (PG2), the active ingredient in dried roots of astragalus membranaceus, ameliorates cancer symptom clusters and improve quality of life (QoL) in patients with metastatic disease by modulating inflammatory cascade, against the background that inflammatory cells including macrophages, dendritic cells (DCs), and cytotoxic T lymphocytes (CTLs) in tumor initiation, metastasis, and progression. Nevertheless, the role of PG2 in the modulation of anticancer immunogenicity and therapeutic response remains relatively underexplored and unclear. Purpose: The present study investigates how and to what extent PG2 modulates cellular and biochemical components of the inflammatory cascade enhance anticancer immunity, and the therapeutic implication of same in patients with lung cancer. Methods and Results: Herein, we demonstrated that PG2 significantly increased the M1/M2 macrophage polarization ratio in non-small cell carcinoma (NSCLC) H441 and H1299 cells. This PG2-induced preferential pharmacologic up-regulation of tumoral M1 population in vitro positively correlated with the downregulation of tumor-promoting IL-6 and IL-10 expression in NSCLC cell-conditioned medium, with concomitant marked inhibition of cell proliferation, clonogenicity and tumorsphere formation. Our ex vivo results, using clinical sample from our NSCLC cohort, demonstrated that PG2 also promoted the functional maturation of DCs with consequent enhancement of T cell-mediated anticancer immune responses. Consistent with the in vitro and ex vivo results, our in vivo studies showed that mice treated with PG2 exhibited significant time-dependent depletion of the tumor-associated M2 population, synergistically enhanced the anti-M2-based anticancer effect of cisplatin, and inhibition of xenograft tumor growth in the NSCLC mice models. Moreover, in the presence of PG2, cisplatin-associated dyscrasia and weight-loss was markedly suppressed. Conclusion: These results do indicate a therapeutically-relevant role for PG2 in modulating the M1/M2 macrophage pool, facilitating DC maturation, and synergistically enhancing the anticancer effect of conventional chemotherapeutic agent, cisplatin; thus laying the foundation for further exploration of the curative relevance of PG2 as surrogate immunotherapy and/or clinical feasibility of its use for maintenance therapy in patients with lung cancer.
Q4: Reviewer #1: In Materials and Methods, the number of patients with other types of cancer (breast, colon, ovarian liver, gastric and brain cancers) must be included.
A4: We thank the reviewer for pointing this omission to us. We have included the number of patients with other types of cancer in our revised manuscript. To determine the ‘reproducibility’ and ‘generalizability’ of these data, we probed for similar trends in patients with breast (n = 4), colon (n = 2), ovarian (n = 2), liver (n = 3), gastric (n = 2) and brain (GBM, n = 2) cancers. Please kindly see Results section, Lines 412-446.
3.6. PG2 up-modulates the CD80+ M1/ CD206+ M2 macrophage ratio and increases the population of CD80+, CD103+, and CD86+ functionally matured dendritic cells, ex vivo
To further characterize the immunogenic potential of PG2, using ELISA and flow cytometry, we comparatively analyzed the ability of PMA, LPS+INF-g or PG2 to modulate the differentiation of the M1/M2 macrophage pool into dendritic cells (DCs) and the functional maturation of immature DCs (iDCs) to mature DCs (mDCs) in cancer patients. We observed that compared to PMA, LPS+INF-g or PG2 induced a 2.4-fold (p<0.01) and 3.7-fold (p<0.01) increase in the number of CD80+ M1 macrophages, respectively, with PG2 inducing 8.5% more CD80+ M1 than LPS+INF-g (Fig. 6A). Conversely, PG2 reduced the number of CD206+ M2 macrophage by 12.1% (p<0.05) and 6.0% (p not significant) compared to PMA and IL-4/IL-13, respectively (Fig. 6B). Since granulocyte-macrophage colony-stimulating factor (GM-CSF) and interleukin-4 (IL-4) trigger monocyte differentiation into DCs, and prevent differentiation to osteoclasts (OCs), as well as induce DC differentiation in the presence of M-CSF and RANK ligand (Hiasa et al., 2009), we compared the effect of GM-CSF+IL4 and PG2 on monocyte differentiation to DCs and DC maturation. We demonstrated that compared to treatment with GM-CSF+IL4 alone, exposure to 16 mg/mL PG2 alone significantly increased the population of CD80+, CD103+ and CD86+ cells by 7% (p<0.05), 8% (p<0.05), and 1% (p = not significant), respectively (Figs. 6C, Figs. D and Figs. 6E), while compared to GM-CSF+IL4 alone, GM-CSF+IL4+PG2 induced more CD80+ (16%, p<0.01), CD103+ (15%, p<0.01) and CD86+ (13%, p<0.05) cells (Figs. 6C, Figs. D and Figs. 6E). These data indicate, at least in part, that PG2 facilitates T-cell activation, M1 differentiation to DCs, and maturation of DCs to mDCs in patients with NSCLC.
To determine the ‘reproducibility’ and ‘generalizability’ of these data, we probed for similar trends in patients with breast (n = 4), colon (n = 2), ovarian (n = 2), liver (n = 3), gastric (n = 2) and brain (GBM, n = 2) cancers. Our results showed that PG2 alone or in combination with GM-CSF+IL4 also enhances the production of functional mDCs, in the breast, colon, ovarian liver, gastric and brain cancers (Fig. 6F and Supplementary Figure S1). Moreover, in light of PG2 immunogenicity in most malignancies, and the observed ability of PG2 to modulate the DCs and macrophage-associated immune landscape, we further investigated if and to what extent treatment with PG2 would affect the activation status and type of T cells in tumor niche. Results of our flow cytometry analysis of dissociated cell suspensions from patient-derived tumor samples indicated that 16 mg/kg PG2 significantly increased the population of CD45+ leucocytes/white blood cells (WBCs) (3.02-fold, p<0.001), which coincidentally seem due to a significant 9.4% expansion of the CD45+CD8+ T cells (p<0.01), with no apparently significant variation in the number of CD4+ T cells in PG2-treated tumor samples compared to their untreated counterparts (Supplementary Figure S2).
Q5: Results on cell migration assays are not in the manuscript. Please, include them.
A5: We sincerely thank the reviewer for this suggestion, however since the present study is particularly focused on the role of PG2 in immunogenicity and not oncogenicity, we humbly request that the reviewer allow us pass on this suggestion.
Q6: Figure 1C and D about the impact on side population and invasion should be in a separate figure or in figure 4.
A6: We sincerely appreciate the reviewer’s comment. Like the reviewer, we also believe that every data is informative, relevant and should be accorded its own space for better appreciation, however, in this stance, we have included these data in Updated Figure 1 because while it (oncogenicity) is not the main focus of the paper, the finding piqued our interest at the early stage of the work, and indeed inform the rest of the study. Therefore, for the preservation of data sequence and scientific logic, we humbly request the erudite reviewer permit the data as is.
Q7: Figure 1D: the picture of M1 and M2 conditioned media at 0h are the same. Please, provide the correct images.
A7: We are sincerely grateful to the reviewer for pointing this grave error. In the spirit of good science and research data integrity, we have corrected this mistake in our Updated Figure 1. Once again, we thank the reviewer for pointing this mistake out.
Q8: Figures 2 and 3. Blue and red colors were used to refer to CD206 and CD80, respectively, in the flow cytometry histograms. However, in the graphs, colors are used in the opposite direction. Please, unify criteria between histograms and bar graphs to avoid confusion.
A8: We are grateful for the reviewer’s insightful comment. As rightly suggested by the reviewer, to avoid confusion, we have now employed consistency in the color coding for the graphs. Please kindly see our updated Figures 2 and 3.
Q9: Figure 4F should be done in the H441 cell line. Besides, the authors must indicate whether that experiment was made in co-culture with M2 macrophages, conditioned media, or in raw parental cells. If not done, this experiment should be done in M2-macrophages conditioned media.
A9: We thank the reviewer for this insightful comment. We apologize for the ambiguity and have now made our description clearly. We have also provided more representative data for the Updated Figure 4F.
Please kindly see our revised Results section, lines 336-360.
3.4. PG2 represses the tumor-promoting effects of anti-inflammatory cytokines and inhibits the NSCLC stem cell-like phenotypes induced by M2-conditioned medium
In separate experiments, we assessed if, how and to what extent PG2-modulated CD80+M1/CD206+M2 ratio affects the CSCs-like phenotypes of NSCLC. Consistent with earlier results, western blot analyses showed that while PG2 up-regulated the expression level of CD80 protein, it down-regulated CD206 protein expression, dose-dependently (Fig. 4A). Furthermore, we demonstrated that PG2 significantly repressed anti-inflammatory IL-10-induced proliferation of tumor-promoting CD206+ cells, compared with the control group (~2.58-fold, p<0.01), and this was akin to the PG2-induced 2.93-fold (p<0.01) decrease in CD206 expression by M2 macrophages cultured in CCCM (Fig. 4B). In validation assays using ELISA, we demonstrated that the significantly enhanced production of IL-10 (2.21-fold, p<0.001) and IL-6 (3.33-fold, p<0.001) by M2 macrophages was efficiently repressed on exposure to 16mg/mL PG as reflected by observed 2.05-fold (p<0.01) and 3.16-fold (p<0.01) decrease in IL-10 and IL-6, respectively (Fig. 4C and Fig. 4D). In the light of these findings, we next examined the effect of PG2 on the CSC-like phenotype, using in vitro NSCLC stem cell models. Interestingly, we observed that in contrast to the cultured M1-MCMs which remained largely adherent, cultured M2-MCMs to a large extent, acquired anchorage-independent phenotype and increased ability to form tumorspheres, which are in vitro CSCs models (Fig. 4E, upper panel). This increased tumorsphere formation ability was more apparent in the H1299 cells. In addition, we observed that co-culture with M1-MCMs induced a moderate decrease in the viability and/or proliferation of the H1299 (I.27-fold, p<0.01) and H441 (I.56-fold, p<0.01) cells, compared to the M2-co-cultured groups (Fig. 4E, lower panel). Of therapeutic relevance, treatment with 0 - 16 mg/mL PG2, significantly suppressed the viability and disrupted the morphological consistency of formed tumorspheres from the M2-co-cultured H1299 cells in a dose-dependent manner (Fig. 4F). These data suggest a putative role for PG2 as an effective inhibitor of M2 macrophage polarization and CSCs-phenotype acquisition, in vitro.
Please also kindly see our updated Figure 4 and its legend, Lines 362-373.
Figure 4. PG2 negatively modulates the secretion of tumor-promoting anti-inflammatory cytokines and inhibits the M2-MCM-induced cancer stem cell-like phenotype of NSCLC cells. (A) Western blot showing a dose-related up-regulation of CD80 and down-regulation of CD206 protein expressions in MDMs after PG2 treatment. (B) Representative histogram of ELISA result showing PG2 significantly inhibited IL-10-enhanced M2 macrophage proliferation and CCCM-cultivated M2 population, compared to the untreated control group. ELISA histogram showing that 16 mg/mL PG2 exposure significantly reversed the CCCM - enhanced (C) IL-10 and (D) IL-6 secretion by M2 cells. (E) Photo (upper panel) and graphical (lower panel) images showing that H441 and H1299 cells co-cultured with M1-MCM remained adherent, while M2-MCM co-cultured cells acquired an adhesion-independent CSC phenotype. (F) PG2 inhibited the viability of tumorspheres grown from M2/H1299 co-culture in a dose-dependent manner. *p < 0.05, **p < 0.01. b-actin served as loading control.
Q10: Figure 5C. Show representative immunohistochemistry images of lung metastases in each treatment group.
A10: We thank the reviewer for this comment. We have also provided more representative data for the Updated Figure 5C.
Please also kindly see our updated Figure 4 and its legend, Lines 402-411.
Figure 5. PG2 suppresses tumorigenicity and metastasis in syngeneic C57BL/6 mice, and potentiates cisplatin anticancer effect in vivo, by modulating inflammation-associated macrophage activity and angiogenesis. (A) Photo images showed the anticancer effect of Cisplatin and/or PG2 in syngeneic C57BL/6 mice inoculated with 1.5x103 LLC1 cells. (B) Graphical representation of the effect of Cisplatin and/or PG2 on the tumor size, tumor weight and body weight in syngeneic C57BL/6 mice inoculated with LLC1 cells. (C) Photo images showed the effect of Cisplatin and/or PG2 on metastasis in syngeneic C57BL/6 mice inoculated with LLC1 cells. (D) Immunofluorescent staining showed that PG2 or Cisplatin alone mildly reduced the expression of NF-kB, CD11b and CD31, while combining Cisplatin with PG2 significantly inhibited their expression in the tissue specimen. Red arrow, liver metastasis; ns, not significant, *p<0.05, **p<0.01, ***p<0.001

Reviewer 2 Report
This manuscript analyzed the effects of PG2 on the cellular and biochemical components of the inflammatory cascade, and also tested its anticancer activities for the treatment of NSCLC. This study is interesting and reveals the novel roles of PG2 in the tumoral M1/M2 macrophage polarization and dendritic cells maturation. The data are clearly presented and support the conclusions.
My only concern of this study is the dosage of PG2 used in this study. The manuscript didn’t explain why the 8mg/mL and 16mg/mL dosages were selected for the treatment. An IC50 curve of PG2 might be necessary to provide some rationale for the dosages.
In addition, effects based on two-dosage treatment cannot be concluded as “dose dependent effects”. Normally, the effects of a high dosage, a medium dosage and a low dosage should be presented.
Author Response
Q1: Reviewer #2: This manuscript analyzed the effects of PG2 on the cellular and biochemical components of the inflammatory cascade, and also tested its anticancer activities for the treatment of NSCLC. This study is interesting and reveals the novel roles of PG2 in the tumoral M1/M2 macrophage polarization and dendritic cells maturation. The data are clearly presented and support the conclusions.
A1: We sincerely thank the reviewer for the time taken to review our work, the encouraging comments and the helpful suggestions given to help us improve the acceptability and appeal of our work.
Q2: Reviewer #2: My only concern of this study is the dosage of PG2 used in this study. The manuscript didn’t explain why the 8mg/mL and 16mg/mL dosages were selected for the treatment. An IC50 curve of PG2 might be necessary to provide some rationale for the dosages.
A2: We thank the reviewer for this important comment. In a study such as ours where the therapeutic agent elicits its effect not based on cytotoxicity to cancerous cells, but on PG2’s ability to induce cell-mediated and/or humoral immune response, selection of a cut-off for high/low concentration may be difficult, however, in our case, though unconventional, we have done this selection of 8mg/mL and 16mg/mL as low and high doses of PG2 based on our observation that compared to the untreated control, PG2 reduced the number of tumorspheres formed by 50% at ~12mg/mL, thus, we used this for dichotomization of PG2 dosage into high (16 mg/mL) or low (8 mg/mL). Please kindly see our Updated Supplementary Figure S3 and its legend. Please see line 720-724.
Supplementary Figure S3. PG2 suppressed the viability of tumorspheres derived from M2/H1229 co-culture system. Compared to the untreated control, PG2 concentration at the number of tumorspheres formed were reduced by 50% is indicated and used for dichotomization of PG2 dosage as high (16 mg/mL) or low (8 mg/mL).
Please kindly see our revised discussion section, lines 506-518.
Moreover, our data showing that PG2 elicits inhibition of NSCLC stem cell-like phenotypes induced by M2-conditioned medium (Figure 4 and Supplementary Figure S3) is therapeutically-relevant, considering CSCs exhibit unlimited self-renewal potential, enhanced ability to drive tumor initiation and disease progression, as well as well-documented complicity in therapy failure and disease recurrence (Sainz et al., 2016; Fan et al., 2014, Gajewski et al., 2013; Ostuni et al., 2015). Beyond being passive residents of the tumor niche, TAMs/MDMs induce autocrine and paracrine signals that enhance CSC survival, maintenance, self-renewal, motility and tumorigenicity (Sainz et al., 2016). In addition, TAM-mediated induction of epithelial-to-mesenchymal transition (EMT) shown to be functionally relevant to the acquisition and maintenance of the CSCs-like phenotype does suggest the critical role of TAMs in the acquisition of a metastatic phenotype by the CSCs (Fan et al., 2014). This highlights the therapeutic relevance of PG2 as a potential CSCs-targeting pharmacological agent and putative component of an effective anticancer therapeutic strategy for patients with NSCLC.
Q3: Reviewer #2: In addition, effects based on two-dosage treatment cannot be concluded as “dose dependent effects”. Normally, the effects of a high dosage, a medium dosage and a low dosage should be presented.
A3: We are very grateful for the reviewer’s comment. As suggested by the reviewer, we have now deleted any reference to dose-dependency where only 2 doses were used in our revised submission.

Reviewer 3 Report
In this work authors explored the role of Astragalus polysaccharide (PG2), the active ingredient in dried roots of astragalus membranaceus, in the modulation of the cellular and biochemical components of the inflammatory cascade, how it enhances anticancer immunity, and the therapeutic implication of patients with lung cancer.
The paper is well written, the research is carefully designed and conducted and an array of data is available including in vitro results, ex vivo data as also in vivo data.
The accumulative data provide novel insights on the role of PG2 as an effective modulator of tumoral M1/M2 macrophage polarization and a potent activator of DC maturation.
Minor suggested corrections: It is advised to alter the first sentence in the conclusion stating that “Taken together, as shown in graphical abstract Figure 7” to “Taken together, these results demonstrate….”
Author Response
Q1: Reviewer #3: In this work authors explored the role of Astragalus polysaccharide (PG2), the active ingredient in dried roots of astragalus membranaceus, in the modulation of the cellular and biochemical components of the inflammatory cascade, how it enhances anticancer immunity, and the therapeutic implication of patients with lung cancer.
A1: We sincerely thank the reviewer for the time taken to review our work, the encouraging comments and the helpful suggestions given to help us improve the acceptability and appeal of our work.
Q2: The paper is well written, the research is carefully designed and conducted, and an array of data is available including in vitro results, ex vivo data as also in vivo data.
A2: We sincerely thank the reviewer for the time taken to review our work, the words of encouragement, and for the important suggestion made to improve the quality of our work.
Q3: The accumulative data provide novel insights on the role of PG2 as an effective modulator of tumoral M1/M2 macrophage polarization and a potent activator of DC maturation.
A3: We sincerely thank the reviewer for the time taken to review our work, our data indicated the therapeutically-relevant role for PG2 in modulating the M1/M2 macrophage pool, facilitating DC maturation, and synergistically enhancing the anticancer effect of conventional chemotherapeutic agent, cisplatin; thus laying the foundation for further exploration of the curative relevance of PG2 as surrogate immunotherapy and/or clinical feasibility of its use for maintenance therapy in patients with lung cancer.
Q4: Minor suggested corrections: It is advised to alter the first sentence in the conclusion stating that “Taken together, as shown in graphical abstract Figure 7” to “Taken together, these results demonstrate….”
A4: We thank the reviewer for this important comment. Please kindly see our revised Conclusion section, lines 562-568.
Taken together, these results demonstrate in graphical abstract Figure 7, our results demonstrate for the first time the novel role of PG2 as an effective modulator of tumoral M1/M2 macrophage polarization and a potent activator of DC maturation. These results provide some mechanistic insight into observed macrophage-mediated PG2-induced immune-based anticancer and chemotherapy-potentiating activity. This study thus lays the foundation for further exploration of the role of PG2 as alternative cancer immunotherapy or maintenance therapy for patients with NSCLC.

Round 2
Reviewer 1 Report
The authors have substantially improved the manuscript and satisfactorily answered all my previous questions and concerns.